# Combination of Hyperbaric Oxygen Therapy and Oral Steroids for the Treatment of Sudden Sensorineural Hearing Loss: Early or Late?

**DOI:** 10.3390/medicina58101421

**Published:** 2022-10-10

**Authors:** Matteo Cavaliere, Pietro De Luca, Alfonso Scarpa, Adriano Maciej Strzalkowski, Massimo Ralli, Matteo Calvanese, Luisa Savignano, Pasquale Viola, Claudia Cassandro, Giuseppe Chiarella, Arianna Di Stadio

**Affiliations:** 1Department of Medicine, Surgery and Dentistry, University of Salerno, 84131 Salerno, Italy; 2Department of Sense Organs, University La Sapienza, 00185 Rome, Italy; 3Unit of Audiology, Regional Centre for Cochlear Implants and ENT Diseases, Department of Experimental and Clinical Medicine, Magna Graecia University, 88100 Catanzaro, Italy; 4Surgical Sciences Department, University of Turin, 10126 Turin, Italy; 5Department G.F. Ingrassia, University of Catania, 95124 Catania, Italy

**Keywords:** hyperbaric oxygen therapy, HBOT, sudden deafness, idiopathic sudden hearing loss, steroid therapy

## Abstract

*Background and Objectives*: Several treatments are available for sudden sensorineural hearing loss (SSNHL), but no studies have compared the different treatments based on the delay from the onset of the disease. Our study aims to compare the effect of hyperbaric oxygen therapy (HBOT), oral steroids (OS) and combination of both therapies (HBOT + OS) for treating SSNHL. *Materials and Methods*: This randomized study analyzed 171 patients with SSNHL. Patients were evaluated by pure tone audiometry test (PTA) at baseline (T0) and 20 days after treatment (T1). Three groups were available HBOT-A-, OS-B- and HBOT + OS-C-. After baseline PTA, patients were randomly assigned to each group. Statistical analysis was performed by one-way ANOVA and Chi-square. *Results*: Patients in the HBOT + OS and HBOT groups improved their auditory function (*p* < 0.05). HBOT was the best choice for treatment when started by 7 days from SSNHL onset, while HBOT + OS in case of late treatment. Profound SNHL recovered equally by HBOT and HBOT + OS (*p* < 0.05). Upsloping SNHL obtained better auditory results by HBOT compared to HBOT + OS (*p* < 0.05). Downsloping and flat SSNHL had the most improvement with HBOT + OS compared to HBOT only (*p* < 0.05). *Conclusions*: Combination of HBOT and OS is a valid treatment for SSNHL both in case of early and late treatment. Combination of HBOT and OS was the choice with the best results in case of treatment started >14 days from symptom onset.

## 1. Introduction

Sudden Sensorineural Hearing Loss (SSNHL) refers to the acute onset (less than 3 days) of perceptive hearing loss with a deficit over 30dB in three contiguous frequencies at least [1]; because of its transitory and spontaneous resolution, the real prevalence SSNHL is unclear but estimated at 5 to 20 per 100,000 individuals. The symptom is commonly unilateral although a bilateral deficit can rarely arise [2]. SSNHL (alone or associate with tinnitus, vertigo, dizziness, and aural fullness) is prevalently observed in population aged 43–53 (77 per 100,000) rather than in people < 18 years (11 per 100,000) [3,4,5]. The pathogenesis of SSNHL is still unclear; however, a multifactorial origin has been hypothesized [6,7]. SSNHL could arise from different disease like viral infections, acoustic tumors, vascular disorders, drug toxicity, autoimmune inner ear disease [6] and psychiatric disorders. Authors have suggested that SSNHL may also be an early symptom of the onset of multiple sclerosis [7]. Despite different possible causes, 90% of patients with SSNHL are discharged without a certain diagnosis. To identify the origin of SSNHL it is necessary to perform several investigations including Pure Tone Audiometry test (PTA), Magnetic Resonance Imaging (MRI) to exclude a retrocochlear pathology, Auditory Brainstem Response (ABR) and specific laboratory tests. The updated international guidelines [1] suggest two therapeutic options: (1) a treatment exclusively based on corticosteroids or (2) the combination of steroids with hyperbaric oxygen therapy (HBOT) [8]. The treatment should start no later than 2 weeks from the onset of SSNHL (early treatment). In case of late treatment (>14 days from symptom onset), HBOT + OS has been suggested as salvage therapy. Finally, the intratympanic steroid therapy could be an alternative option in those patients who present incomplete recovery of their hearing function 2 to 6 weeks after SSNHL onset [8]. The time of initial treatment seems to be the key to obtain the best results in term of recovery of the hearing function; in fact, most of the recovery occurs by two weeks from SSNHL onset, sometimes spontaneously (35–65% of cases) [9]. Some controversies are still open about the best treatment for SSNHL; only few studies compared different treatments at different time of the initial treatment (early vs. late) [10,11]. Our study aims at evaluating the efficacy, in term of hearing recovery, of HBOT, oral steroids (OS) or combination of both, using each treatment at different times from symptom onset to identify the best (early and/or late) treatment in patients suffering from SSNHL.

## 2. Materials and Methods

One-hundred seventy-one patients affected by SSNHL, who presented to the Otolaryngology department of our tertiary referral center from February 2016 to December 2019 were analyzed. All patients included in the study signed a written consent. Inclusion and exclusion criteria were defined to include/exclude the patients. Inclusion criteria were age > 18 years, onset of SSNHL in the last 30 days, unilateral and/or bilateral symptom(s), unknown cause of hearing loss, no fluctuations in hearing loss, no previous otologic surgery in the ear affected from SSNHL, no previous cancer treatment, normal function of Eustachian tube. Exclusion criteria were age < 18 years, known cause of hearing loss, persistent SSNHL > 31 days, previous history of cancer, hypertension not under control, untreated diabetes, history of stroke, current or history of neurologic and/or psychiatric disorders. All patients were initially screened as follow: otolaryngology consultation, routine blood tests, electrocardiogram (ECG), measurement of blood pressure and brain MRI with and without contrast, to exclude a retrocochlear pathology.

Only the patients meeting inclusion/exclusion criteria were considered for inclusion in the study. Three groups were defined. Group A, in which patients were treated by HBOT exclusively (HBOT)—one session per day from Monday to Friday at 2.5 ATA with 90 min per session (time of the whole HBO session), for a variable total number of sessions for 15 days (10 sessions total); Group B, in which patients were exclusively treated by oral steroid therapy (OS)—oral prednisone 1 mg/kg per day (for a maximum dose of 60 mg per day) for 12–14 consecutive days; Group C, which included patients treated by combining HBOT and OS. All patients performed a PTA at T0 (pre-treatment) and T1 (twenty days after treatment). The following information were collected: age, sex, comorbidities, time of initial treatment.

### 2.1. Randomization and Allocation

The 171 participants were randomized into 1 of the 3 groups (HBOT, OS, HBOT + OS) using the following procedures. A block randomization was used, to have a balanced number of participants in each group (45–55% range of assignment ratio). A free randomization software (Random Allocation Software 1.0) provided the randomization sequence. Allocation concealment was achieved by using the sequentially numbered, opaque, sealed envelopes method.

### 2.2. Auditory Investigation

The patients were tested by earphones in a silent cabin. A pulse-tone was emitted by the speaker on the side of the ear that had to be tested, and a white-noise sound (masquerading sound) was sent by an insert located on the opposite side. The sound stimulation started from 10 dB, with increases of 10 dB and decreases of 5 dB, to confirm the sound perception. The following frequencies were tested: 250, 500, 1000, 2000, 4000, 6000 and 8000 Hz. Pure tone average (PTAv) was calculated by summing the results of PTA thresholds at 500, 1000, 2000 and 4000 Hz and dividing the total by four. Once the thresholds were identified, the SSNHL was classified as follow: Upsloping (hearing loss affecting 250 and 500 Hz more); Flat (<20 dB difference between the highest and the lowest threshold); Downsloping (hearing loss affecting 4000 and 8000 Hz more; Profound (thresholds of 90 dB or more in each test frequency).

### 2.3. Data Analysis

At first, one-way ANOVA was performed to compare patients’ PTAv before and after treatment in the three groups. Ad-hoc Bonferroni-Holms (BH) test was used. The auditory recovery based on the different SSNHL thresholds and treatment used was compared using Chi-square. Then the effect of the starting-time of the therapy on the recovery of auditory function was analyzed using One-way ANOVA and Ad-hoc BH test. For this analysis the patients were grouped as follow: <7 days, 8–14 days, and >14 days. Thirdly, the impact of the age on the auditory recovery after the different treatments was analyzed by Chi-square, classifying the age as follow: <50-year-old and >50-year-old. In the end, one-way ANOVA was performed to compare the effect of the therapy between women and men, analyzing the different treatments and the auditory thresholds; Ad-Hoc BH was performed. *p* was considered significant <0.05. The analyses were performed using STATA^®^.

## 3. Results

Group HBOT (A) and group OS (B) included 60 and 55 patients, respectively, and group HBOT + 128 OS (C) was consisted of 56 patients (Table 1). No short- or long-term post-treatment complications were observed.

### 3.1. Comparison among Treatments: PTAv and SSNHL Type

All patients improved their PTAv comparing the scores before (T0) and after treatment (T1) independently from the treatment used (ANOVA: *p* < 0.05) (Figure 1); in particular patients who underwent combination of treatments (HBOT + OS) presented a statistically significant improvement better than the ones in group HBOT [*p* < 0.05 (BH)] and OS [*p* < 0.05 (BH)]. Patient who underwent HBOT obtained a better recovery compared to the ones treated by OS only (BH: *p* < 0.05) (Figure 1). Group HBOT and HBOT + OS always showed statistically significant variations before and after treatments, respectively *p* < 0.05 and <0.05. Observation of SSNHL type showed that patients with profound type recovered their auditory thresholds without statistically significant differences both using HBOT and HBOT + OS (*p* = 0.08). Upsloping type showed better results using HBOT compared to HBOT + OS (*p* < 0.05). Downsloping and flat type had better results by using HBOT + OS compared to HBOT only (*p* < 0.05) (Figure 2). The PTAv of patient treated by OS, despite the improvement of the scores after treatment, never reached statistically significant values. In addition, the absence of a statistically significant values was observed all cases independently from the SSNHL types.

### 3.2. Comparison among the Times of Initial Treatment

Patients who started the therapy within 7 days from SSNHL onset presented a statistically significant recovery of their PTAv after treatment (ANOVA: *p* < 0.05) if they were treated by HBOT (HB: *p* < 0.05) or HBOT + OS (HB: *p* < 0.05), but not if treated by OS (HB: *p* = 0.08). When the treatment started 8–14 days from symptom onset, the recovery was not statistically significant (ANOVA: *p* = 0.07), except in case of patients treated by HBOT (*p* < 0.05). However, the recovery of PTAv was observed in all cases, despite not statistically significant; patients treated by HBOT + OS and HBOT presented better PTAv compared to patients who underwent OS. If the patients were treated >14 days after SSNHL onset, the improvement of PTAv was never statistically significant (ANOVA: *p* = 0.08). However, the recovery of PTAv was better in group HBOT + OS than in groups HBOT and OS. (Figure 3).

### 3.3. Comparison among Patients’ Age

We observed that HBOT was better than HBOT + OS to improve PTAv in patients < 50 years old (*p* < 0.05), while in subjects over 50-years the association of therapies (HBOT + OS) was the best method to recover the auditory capacity (*p* < 0.05). The recovery in OS group was always not statically significant independently from patients’ age. 

### 3.4. Gender Comparison

Women recovered better (ANOVA: *p* < 0.05) compared to men (women’s PTAv average showed lower scores than men’s). The better results were achieved by using HBOT + OS (HB: *p* = 0.001) compared to HBOT or OS and HBOT (HB: *p* < 0.05) versus OS. HBOT + OS (HB: *p* = 0.04) was better than HBOT only.

## 4. Discussion

Overall, our study shows that HBOT is better than OS for treating SSNHL, especially when the treatment is started within 14 days from the symptom onset. In our series, OS treatment, although allowed a partial recover of the hearing capacities, never showed better results compared to HBOT alone or association of therapy (HBOT + OS). HBOT has been used to treat inner ear diseases since early 1970s, because of its benefic effect on cochlear perfusion. In fact, cochlear hypoperfusion is one of the causes of SSNHL [12]; the rationale behind the use of HBOT, in which an individual breathes nearly 100% oxygen in a hyperbaric chamber pressurized at higher than the pressure at sea-level (1 ATA) [13], seems to be quite strong to affirm that this treatment should be always considered in all cases of inner ear disorders related to deficit/alteration of perfusion. Furthermore, because of its safety and efficacy, HBOT has been suggested as first-line treatment for idiopathic SSNHL by the American Academy of Otolaryngology-Head and Neck Surgery [14]. Our results confirm the efficacy of HBOT (alone or associated with steroid treatment) as early treatment of SSNHL as previously demonstrated by Bennet et al. study [15]; in their study the authors showed an improvement of PTAv after uses of HBOT and we observed the same benefic effect of the treatment in our sample. We also observed that people with profound SSNHL recovered the normal auditory thresholds both if they treated by HBOT only and by HBOT + OS. Patients with upsloping SSNHL reached better results using HBOT and the ones with downsloping and flat SSNHL obtained better results using HBOT + OS. Because of the different frequential distribution of hair cells into the cochlea, we speculated that the efficacy of the different therapies could be related to the position of the cells in the cochlea turns [16]. Upsloping SSNHL indicates the loss of low frequency (250–500 Hz) which are tonotopically located in the apex of the cochlea; the HBOT increasing both O_2_ blood concentration and pressure by ameliorating the perfusion in the apex of the cochlea; this allows the recovery of the lower frequencies thanks to the restoring of the function of the hair cells in this area. In case of flat and downsloping SSNHL are the base and the middle turn of the cochlea with their hair cells that suffering, so the combination between HBOT and OS allows the recovery of the auditory function both ameliorating the vascular support and reducing the inflammation. The hair cells in the middle and basal turn of the cochlea are numerically more than the ones in the apex [16], so in case of cellular sufferance these cells produce pro-inflammatory cytokines that ulteriorly worsening the inner ear environment inducing cells apoptosis [17]. Steroids can quickly stop this phenome. 

The patients treated by OS, despite improved their auditory thresholds, never reached values statistically significant. Our observations are supported by Choi et al. [18] study, in which the authors, evaluating 38 patients who were initially treated with HBOT, noted that these people presented a significantly higher hearing improvement compared with a control group treated by OS only; the improvement was particularly evident in the low frequency (0.5 kHz, 1 kHz, 2 kHz; *p* < 0.05). Alimoglu et al. [19] obtained similar results comparing oral steroid therapy, intratympanic steroid therapy, HBOT therapy alone, and HBOT combined with oral steroid therapy. The better recovery of the low frequencies can be explained looking at the tonotopic location of hearing frequencies into the cochlea; the low frequencies are in the apex of the structure [20], which is extremely susceptible to the variation of systemic pressure and hypoperfusion. HBOT improves the systemic oxygenation and consequently the concentration of O_2_ into the inner ear artery; the O_2_ concentration higher than normal allows a better perfusion of the apex of cochlea with consequent recovery of the auditory function. The improvement of oxygenation could also explain why both women and men recovered from SSNHL using HBOT, despite some differences in their different cardiovascular apparatus [21]; the therapy was in all cases a valid solution to recover the hearing capacity, despite the gender cardiovascular differences -which could explain the better results observed in women. The vascular system of the inner ear could the key to explain why the subject over 50 ies particularly benefit of HBOT associated with OS. The vascular apparatus of the inner ear tends to decrease in caliber size due to arteriosclerosis [22], so the combination of HBOT with OS, which increases systemic pressure and determines peripheral vasodilation [23] could explain why this treatment was better than the single HBOT treatment in patients over 50 years old; in fact, the vasodilation affects also arteriosclerotic vessels and allows a better perfusion of the cochlea. The positive effect of combining HBOT with OS as SSNHL treatment has been widely confirmed by other authors [24,25], and our results are an ulterior confirmation of the validity of this treatment. The early treatment (<14 days) by HBOT allowed to obtain the improvement of PTAv also in those cases in which patients presented extremely poor auditory thresholds (profound and flat SSNHL); our findings were similar to the ones observed by other authors [26,27]. Based on the results of our study and supported by the literature [28,29,30], we recommend the early treatment of SSNHL (<7 and by 14 days) using HBOT as first line treatment, in association with OS. After 14 days HBOT (alone or combined with OS) can be used to support the recovery the hearing capacity as widely confirmed in literature [28,29,30].

Future studies should be addressed to investigate the changes in the speech perception test (SPT) before and after treatment; this info could reinforce the efficacy of our results or evidence the limitation of these treatments to preserve the good quantity and quality of hair cells. In fact, to allow correct identification of the words both hair cells and spiral ganglions functions have to be quite good [20].

This study presents some limitations. As first, the patients underwent audiological screening only and no details were available on vestibular function. Second, patients did not perform speech test before and after treatment, so the results can only evaluate the effect of therapy on the cochlea, but they lack in info about the integration in the brain. Thirdly, in some cases, information about auditory functions before the episode of SSNHL were not available. In addition, despite randomization groups OS and HBOT + OS had a different in age greater than 20 years in age and greater than 10 dB in pre-treatment that could impact on the results. Finally, our study lacks the control group (without treatment) and this could be relevant because 38–65% of patients can spontaneously recover and it could be a possible confounder in our study.

## 5. Conclusions

Our study seems to confirm the efficacy of combination of HBOT and OS in the treatment of SSNHL, especially if the treatment starts within 14 days from symptom onset. Moreover, HBOT + OS should be considered as first line treatment and can also be considered a valid option for late treatment of SSNHL.

## Figures and Tables

**Figure 1 medicina-58-01421-f001:**
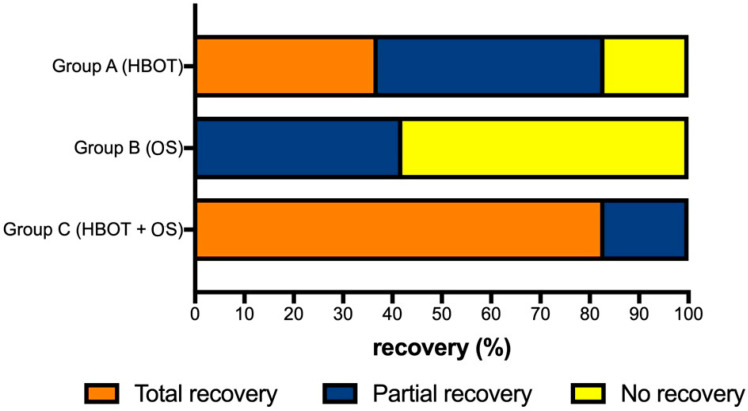
Comparison of hearing recovery among the groups.

**Figure 2 medicina-58-01421-f002:**
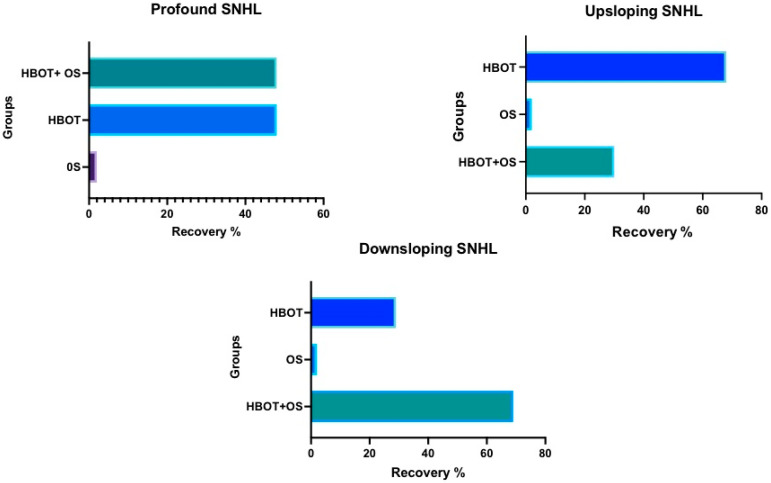
The graphs show the prevalence of recovering comparing the groups. Profound SNHL recovers same comparing HBOT alone or combined with OS; the Upsloping SNHL recovers better by HBOT alone than HBOT + OS; finally, Downsloping SNHL recovers better using HBOT + OS compared to HBOT only. In all cases, OS as single treatment is not sufficient to recover.

**Figure 3 medicina-58-01421-f003:**
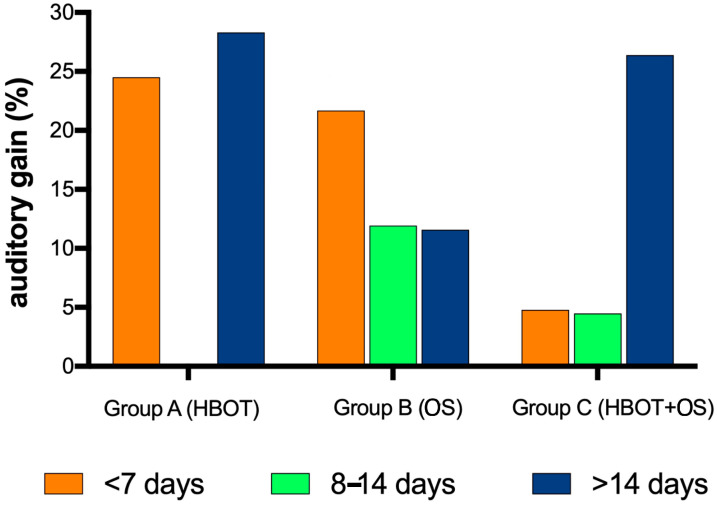
The graph shows the differences in the recovery at different time points among the three groups. HBOT and HBOT + OS allow to improve the hearing functions even after more than 15 days from the hearing loss onset.

**Table 1 medicina-58-01421-t001:** Demographic characteristics of our sample.

	HBOT	OS	HBOT + OS
No. of patients	60	55	56
Age (mean ± SD)	55.7 ± 14.17	67.7 ± 9.35	44.1 ± 13.81
Gender F/M	29/31	26/29	25/31
Side (right/left/bilateral)	31/29/0	29/26/0	30/25/1
Pre-treatment PTA (dB)	57.79 ± 25.5	66.25 ± 19.73	55.9 ± 23.89
Audiogram type (%)			
Upsloping	14.8	0	13.9
Flat	44.4	50	38.9
Downsloping	31.5	41.7	33.3
Profound	9.3	8.3	13.9

## Data Availability

Not applicable.

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
