# Peer review of "Combination of Hyperbaric Oxygen Therapy and Oral Steroids for the Treatment of Sudden Sensorineural Hearing Loss: Early or Late?"

_medicina, 2022, doi:10.3390/medicina58101421_

Round 1

Reviewer 1 Report

Cavaliere and colleagues investigated and compared the effects of hyperbaric oxygen therapy (HBOT), oral steroids (OS) and combination of both therapies (HBOT + OS) on the sudden sensorineural hearing loss (SSNHL). The authors clearly showed that the HBOT or the combination has a better treatment outcome than OS, and HBOT+OS combination should be considered as the first line treatment for SSNHL whenever possible. This is a very neat study and should be valuable for developing a standard for treatment of SSNHL. I only a few minor points below.

1.      Line 32, “Combination ofHBOT and OS…” should be “Combination of HBOT and OS…”

2.      Line 57, There should be a space between (HBOT) and [8]

3.      Line 128, The sentence can be revised to “Group HBOT (A) and group OS (B) included 60 and 55 patients, respectively, and group HBOT + 128 OS (C) was consisted of 56 patients (Table 1)”

4.      Line 177, should be p=0.08

5.      Line 193, should be <50 years old

6.      Line 206 and 207, the “thera-py.” should be deleted.

7.      It would be nice to give a little explanation/definition about downsloping/upsloping type in the first place.

Reviewer 2 Report

Title - Clearly defines the work.

Abstract - It is well elaborated, succinct and summarizes the article.

Introduction - It is appropriate to the topic

Materials and methods - Clearly explains the methodology used, characterizes the sample of participants.

Comment/suggestion for future work: Some of these patients who recover hearing thresholds have complaints of difficulty in understanding speech, so the work could have been more valued if vocal audiometry and otoacoustic emissions had been included in the two assessment times (T0) and (T1).

Data analysis - the statistical methodology used for the evaluation/comparison of the results obtained in the different evaluation conditions was clearly explained.

Discussion - Performed in a clear and enlightening way, demonstrating the benefits of using this treatment, the benefit to use the combination of hyperbaric oxygen therapy and oral steroids in early and late treatment. The study presents the influence of age on the effectiveness of treatment and the influence of gender on the results obtained, and the benefit of early start of treatment.

References - The list of references is adequate, although a little old (in 35 references only one work from 2019 and another from 2020 appears)

Graphs, figures and tables - They are very didactic and complement the written text.
